# Treadmill belt accelerations may not accurately replicate kinematic responses to tripping on an obstacle in older people

**Dayeon C. Jung[1,2], Daina L. Sturnieks[1,2], Kirsty A. McDonald[3], Patrick Y. H. Song[1,4], Michael K. Davis[1,5], Stephen R. Lord[1,6], Yoshiro Okubo[1,6]***

**1** Falls, Balance and Injury Research Centre, Neuroscience Research Australia, Sydney, Australia, **2** School of Medical Sciences, University of New South Wales, Sydney, Australia, **3** School of Health Sciences, University of New South Wales, Sydney, Australia, **4** School of Medicine, University of New South Wales, Sydney, Australia, **5** College of Health Sciences, University of Delaware, Newark, DE, United States of America, **6** School of Population Health, University of New South Wales, Sydney, Australia

* y.okubo@neura.edu.au

## Abstract

### Background

Treadmill belt perturbations have high clinical feasibility for use in perturbation-based training in older people, but their kinematic validity is unclear. This study examined the kinematic validity of treadmill belt accelerations as a surrogate for overground walkway trips during gait in older people.

### Methods

Thirty-eight community-dwelling older people were exposed to two unilateral belt accelerations (8 m s$^{-2}$) whilst walking on a split-belt treadmill and two trips induced by a 14 cm tripboard whilst walking on a walkway with condition presentation randomised. Anteroposterior margin of stability (MoS), number of falls, and trunk and lower limb kinematics were quantified for the step prior and five recovery steps following the treadmill perturbations and the walkway trips which elicited elevating and lowering strategies.

### Findings

Rates of falls following the treadmill accelerations and walkway trips were 0% and 13.1%, respectively. MoS was similar during the first recovery step ($P>0.05$) but less negative during subsequent recovery steps following treadmill belt accelerations than walkway trips ($P<0.01$) regardless of recovery strategy. Excluding the first recovery step in the lowering strategy, recovery step lengths, toe clearance, maximum trunk, hip and knee angles ($P<0.05$) were smaller during recovery on the treadmill compared to the walkway.

### Interpretation

Destabilisation by treadmill belt accelerations quickly dissipated after only one recovery step but continued for multiple recovery steps following walkway trips. Smaller trunk

**Data Availability Statement:** The dataset underlying this study cannot be made publicly available for ethical reasons. It is stored at

Neuroscience Research Australia, with Yoshiro Okubo (y.okubo@neura.edu.au), Daina Sturnieks (d.sturnieks@neura.edu.au), and Stephen Lord (s.lord@neura.edu.au) serving as the data custodians. Any requests for data access should be directed to them and will be evaluated by the University of New South Wales Human Research Ethics Committee (humanethics@unsw.edu.au) before any data is released. A request for data access should include but not limited to: Purpose of the request, affiliation, contact details, research ethics approval (if applicable), data security plan, intended use and time frame.

**Funding:** The author(s) received no specific funding for this work.

**Competing interests:** The authors have declared that no competing interests exist.

displacement, step lengths, toe clearance and no falls on the treadmill indicate treadmill belt accelerations may not accurately simulate the biomechanical challenge of obstacle-induced trips in older people.

## Introduction

Tripping over an object or uneven surface while walking is a leading cause of falls in community-dwelling older people [1]. Studies have reported that older people demonstrate reduced foot clearance and ability to clear obstacles compared to young adults [2–4]. The risk of foot collision with an obstacle is increased by impaired depth perception and poorer dual tasking abilities [5]. When a trip occurs, the body's centre of mass (CoM) accelerates forwards and downwards beyond a stalled base of support (BoS) and requires adequate reactive stepping to establish a new BoS to regain stability [6, 7]. This is achieved by elevating the obstructed foot over the obstacle (i.e., an elevating strategy) or lowering the obstructed foot before the obstacle and subsequent elevation of the contralateral foot to clear the obstacle (i.e., a lowering strategy) [8]. Age-related reduction in lower limb proprioception [9] and muscle function can hinder effective reactive stepping for trip recovery [7, 10, 11]. However, older people can improve their trip recovery responses through task-specific training that involves repeated exposure to simulated trips [12–14]. Similarity between surrogate trips and real-life trips may increase the generalisability and chances of recovery from unexpected trip scenarios [15]. Preliminary evidence suggests that task-specific training effects may extend beyond the laboratory and reduce trip-related falls in daily life [16, 17].

Various trip simulation methods have been used in perturbation-based fall prevention studies. Forward reactive steps can be induced while standing with tether-release [18], waist-cable pulls [19] or with a sudden start of a treadmill belt with or without an obstacle [13, 15, 20]. Given that most trips occur during gait [21], which increases the complexity of maintaining dynamic stability [22], the ecological validity of these standing perturbation methods may not be high. Trips during gait have been simulated with 5–14 cm high trip-boards [8, 14, 23] or blocks [6] that suddenly rise from a walkway to obstruct the foot during the swing phase. However, such perturbation walkways require a large space and a ceiling harness track which limits their clinical application. Conversely, treadmills capable of rapid accelerations are clinically advantageous given the compact setup and precise perturbation parameters to evoke reactive gait responses during continuous walking [24]. Compared to other methods involving block obstacles on the treadmill surface [25, 26], or backward ankle-cable pulls [27, 28], treadmills that can induce rapid belt accelerations during gait [14, 29] are easier to administer. Treadmill belt accelerations have been associated with reduced dynamic stability during the first recovery step [14] and sudden forward trunk flexion [30], which are also seen following obstacle-induced trip responses [25, 28]. However, since treadmill belt accelerations do not obstruct the swing foot, the stepping kinematics may be different from genuine trip responses which involve elevating or lowering the tripped foot to clear the obstacle [8]. No previous studies have directly contrasted the kinematic responses induced by treadmill belt accelerations and to those induced by obstacle-induced trips while walking on a walkway [15].

This study aimed to examine the similarity between kinematic responses induced by treadmill belt accelerations and those induced by obstacle-trips which include lowering and elevating strategies during gait in older people. Using maximum acceleration properties we considered to be safe for older adults, we tested the following hypotheses: (1) While treadmill

 

belt accelerations may initially destabilise balance (i.e., a negative margin of stability [MoS]) to the same magnitude as walkway trips (no statistically significant difference), such destabilisation would quickly dissipate following the first recovery step only on the treadmill [14]. (2) Maximum trunk flexion following treadmill and walkway perturbations would be similar, but (3) step kinematics (toe clearance and step lengths) and lower-limb joint angles (maximum hip and knee flexion angles) following treadmill belt accelerations would be smaller compared to walkway trips [15]. Providing answers to these questions would contribute to the development of biomechanically valid task-specific assessment and training protocols to reduce trip-related falls in older people.

## Methods

### Study design

This experimental study was a planned secondary investigation of a cross-over randomised controlled trial [14]. The study conformed to the Declaration of Helsinki and was approved by the University of New South Wales Human Research Ethics Committee (HC16227).

### Participants

Participants were recruited via the Neuroscience Research Australia Research Volunteer Registry in July 2019 to March 2020. Eligibility criteria included adults aged 65 years or above who were living independently, able to walk unaided for 20 min or more and without osteoporosis or a neurological condition. All participants provided written informed consent.

### Experimental protocol

Initially, participants were assessed for body weight, height and leg dominance ('which leg would you kick a ball with?') and asked to walk at usual gait speed along an 8 m walkway containing a 5.7 m GAITRite pressure sensitive mat (CIR Systems, Franklin, NJ, USA) three times to quantify usual step length, cadence and velocity. Participants were fitted with a full body harness, protective guards on knees, shins, foot instep and toes (placed over the shoes). Thirty-nine 14 mm diameter retro-reflective markers were affixed to participants' anatomical landmarks in accordance with the Plug-in-Gait full-body marker set [31]. Participants started with either the walkway (*n* = 19) or treadmill condition (*n* = 19) based on a coin flip and completed both conditions.

### Walkway

Participants familiarised themselves (3 min) with walking on a custom-built 10 m wooden walkway that concealed a trip-board (height: 14 cm, width: 50 cm) (S1 Fig in S1 File) [23] and interfaced with an eight infrared camera motion capture system (100 Hz; Vantage, Vicon, Oxford, UK). To ensure participants consistently walked at ~90% of their usual walking speed, stepping tiles were placed along the walkway at 95% of participants' usual step length and they were instructed to step on these tiles in time with a metronome set at 95% of their usual cadence. Participants were informed that they may experience a hazard anywhere and at any time whilst walking on the walkway but to try to continue walking normally. We administered five unperturbed walking trials (each for 10 m) followed by two semi-random trip trials separated by at least one unperturbed walking trial to ensure no predictive behaviours. In the first perturbation trial, the trip-board obstructed the non-dominant left leg during the mid- to late-swing phase (≥50%) on the middle of the walkway (Fig 1B). In the second perturbation trial, the trip was delivered to the left leg during the early- to mid-swing phase at a different location

 

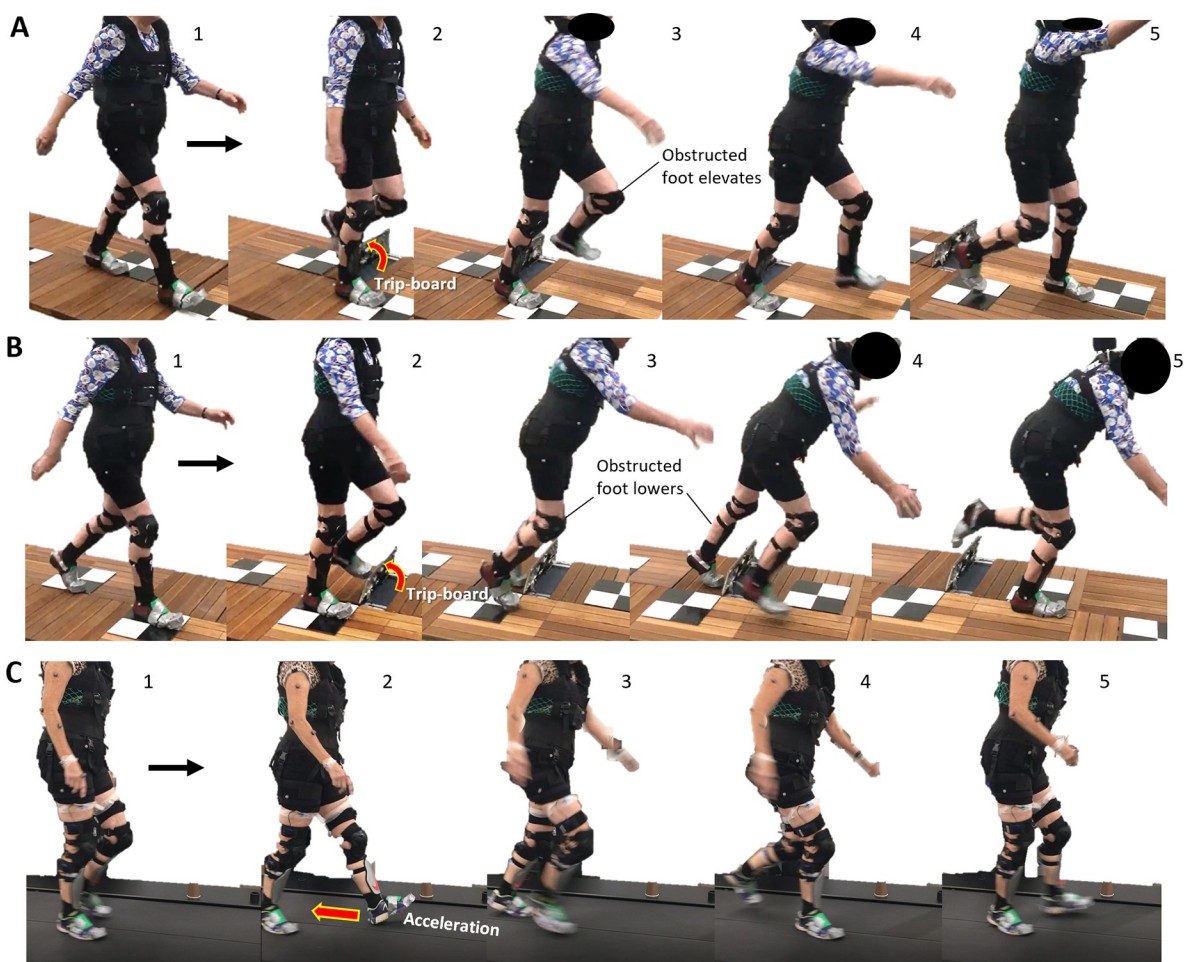

**Fig 1.** Images of trips on the walkway that induced an (A) elevating, (B) lowering recovery strategies, and (C) treadmill accelerations.

on the walkway (Fig 1A). Trips were induced using a 14 cm height tripping board that flipped up from the walkway when the foot passed a hidden sensor at mid-swing.

## Treadmill

Participants practised walking (3 min) on a split-belt treadmill (M-Gait, Motekforce Link, Amsterdam, NL) (without handrails) and interfaced with an eight infrared camera motion capture system (100 Hz; Bonita, Vicon, Oxford, UK). The treadmill belt velocity was fixed to 90% of participants' usual walking speed (as assessed on the GAITRite mat). Participants were informed that they may experience a hazard at any time whilst walking on the treadmill but to try to continue walking normally. We administered five unperturbed walking trials (each lasting 30 s) followed by two semi-randomly presented perturbation trials separated by at least one unperturbed walking trial. In both perturbation trials, the non-dominant left leg was perturbed via treadmill belt acceleration at 8 m s$^{-2}$ to a maximum of 200% of usual walking speed for 30% of stride time [14] (Fig 1C; S2 Fig in S1 File). Perturbations were triggered when the hallux motion capture marker of the ipsilateral swing foot as it passed the hallux marker of the contralateral limb in the sagittal plane [29]. Our perturbation magnitude was determined following a pilot session with a middle-aged participant who reported muscle strain with a

**Table 1. Key participant demographics and baseline spatiotemporal parameters of gait stratified by gender.** Data are mean (standard deviation).

|  | Total sample | Female | Male |
|---|---|---|---|
|  | (*n* = 38) | (*n* = 21) | (*n* = 17) |
| Age (years) | 73.6 (4.7) | 73.5 (4.6) | 73.6 (4.8) |
| Body weight (kg) | 74.3 (13.1) | 66.1 (9.6) | 84.4 (9.3) |
| Body height (m) | 1.69 (0.10) | 1.62 (0.06) | 1.77 (0.06) |
| Step length (cm) | 64.78 (10.08) | 60.86 (7.73) | 69.07 (10.94) |
| Cadence (steps min$^{-1}$) | 107.5 (9.0) | 111.1 (8.5) | 103.0 (7.6) |
| Gait speed (m s$^{-1}$) | 1.16 (0.21) | 1.13 (0.19) | 1.19 (0.24) |

treadmill belt acceleration at 12 m s$^{-2}$ to a maximum of 200% of usual walking speed for 30% of stride time.

## Outcome measures

**Recovery strategies and falls.** Trip trials on the walkway were classified into the elevating (i.e., the obstructed foot elevated and cleared the obstacle) or lowering (i.e., the obstructed foot lowered and the contralateral foot cleared the obstacle) recovery strategies. A fall was defined when a harness load cell recorded >30% of participants' body weight [32].

**Kinematics.** Custom MATLAB scripts (R2019b, MathWorks, MA, USA) were used to calculate kinematic variables for one previous (Pre) and five recovery steps (Post1-5) after trip-onset. On the walkway, regardless of the recovery strategy, the first foot-strike of the obstructed non-dominant left foot defined the end of Post1 and start of Post2. On the treadmill, the first foot-strike of the unperturbed dominant right leg defined the end of Post1 and start of Post2. Dynamic stability was quantified by the anteroposterior (AP) MoS at the foot-strike of each step [28, 33]. Positive MoS signifies that an individual's body is in a state of stable balance (extrapolated centre of mass (*xCoM*) is within the BoS), while negative MoS indicates body instability (*xCoM* is outside the BoS).

The magnitude of a perturbation was quantified as the *xCoM* position at each foot-strike relative to the position of the ankle marker at the previous foot-strike (cm). A larger *xCoM* value signifies a greater anterior progression of *xCoM* during the step, thus indicating a greater magnitude of perturbation. The maximum AP velocity of the *CoM* (m s$^{-1}$) during each step was calculated relative to the treadmill belt velocity or ground. Step length (cm), maximum toe clearance (cm), and maximum trunk, hip and knee angles (°) during each step were also derived. See Appendices in S1 File for more details.

For the fifth and sixth unperturbed walking trials, average MoS (cm), gait speed (m s$^{-1}$), cadence (steps min$^{-1}$) and step length (cm) were calculated, and the results pooled for analysis of unperturbed gait characteristics.

## Statistical analysis

A generalised linear mixed-effects model (GLMM) with robust estimation was used to examine the differences in responses on the treadmill and walkway. The model fit was inspected using Q–Q plots of the residuals and data were log-transformed to improve model fit. Unperturbed gait characteristics (MoS, gait speed, cadence, step length) between conditions (treadmill vs. walkway) were examined with the GLMM, adjusting for trial number (fifth/sixth trials) and order of condition (treadmill or walkway first) as fixed factors and participant number as a random factor. Two GLMMs were used to compare continuous kinematic outcomes

between conditions (1. treadmill vs. lowering strategy, 2. treadmill vs. elevating strategy) for each step (Pre, Post1-5) adjusting for trial number (first/second trials), order of condition as fixed factors and participant number as a random factor. To control for the family-wise error rate, unadjusted $P$ values obtained from the GLMM were corrected with the Holm-Bonferroni method [34]. Statistical analyses were conducted using IBM SPSS Statistics 26 (IBM Corp., Armonk, NY). $P<0.05$ was considered statistically significant.

## Results

The 38 older participants (21 females and 17 males) were right limb dominant, with average usual step length 64.78±10.08 cm, cadence 107.5±9.0 steps min$^{-1}$ and gait speed 1.16 ±0.21 m s$^{-1}$. Participant characteristics are stratified by gender in Table 1.

Participants walked with similar MoS and gait speed during unperturbed gait on the walkway and treadmill ($P>0.05$) but with shorter step lengths on the treadmill (57.4±9.6 cm) compared to the walkway (63.2±9.6 cm) and with higher cadence on the treadmill (112.2±8.5 steps min$^{-1}$) compared to the walkway (108.0±10.3 steps min$^{-1}$) ($P<0.001$). Fall rates following walkway trips were 13.1%, with the elevating and lowering strategies eliciting 22.7% and 8.6% rates of falls, respectively. Treadmill accelerations resulted in a 0% fall rate (n/person-trial) (note, the elevating and lowering strategies were not applicable).

Compared to the treadmill, MoS was more negative during both the walkway elevating (Post1-3) and lowering (Post2-5) strategies ($P<0.01$) (Fig 2A). Compared to the treadmill, xCoM was more anterior during the walkway elevating and lowering strategies in all steps, except for Post1 during which xCoM was more posterior during the walkway lowering strategy compared to the treadmill ($P<0.05$) (Fig 2B). Maximum CoM AP velocities during both walkway strategies were greater than those on the treadmill during all steps ($P<0.05$) (Fig 2C). Compared to the treadmill, step length was greater during the walkway elevating (all steps) and lowering (Pre, Post2-3) strategies ($P<0.05$) (Fig 2D), except for Post1, which was lower during the walkway lowering strategy than the treadmill (31.7±12.0 vs. 55.0±9.5 cm, respectively, $P<0.001$). Maximum toe clearance was higher during both walkway elevating (Post1-3, Post5) and lowering (Post3-5) strategies compared to the treadmill ($P<0.001$) (Fig 2E).

Fig 3 displays the time normalised trunk angles during Pre and Post1-5 steps on the treadmill and during walkway elevating and lowering strategies to illustrate the trunk angle displacements over time. The increase in trunk angle on the treadmill was higher than those on the walkway for both strategies. The maximum trunk angle was greater during the walkway elevating (Pre, Post1-3) and lowering (Post2-5) strategies compared to the treadmill except for Post1, during which maximal trunk angle was lower during the walkway lowering strategy compared to the treadmill ($P<0.001$) (Figs 3 and 4A). Compared to the treadmill, the maximum hip angles were greater during the walkway elevating (Post1-3) and lowering (Post2-4) strategies except for Post1, during which the maximum hip angle was lower during the walkway lowering strategy than on the treadmill ($P<0.01$) (Fig 4B). Similarly, the maximum knee angles were greater during the walkway elevating (Pre, Post1-4) and lowering (Post2-4) strategies compared to the treadmill, except during walkway Post1 in which the maximum knee angle was lower than the treadmill ($P<0.05$) (Fig 4C).

## Discussion

### Magnitude of destabilisation

Supporting our first hypothesis, both treadmill and walkway perturbations delivered initial disturbances (i.e., MoS during the first recovery step) to a similar extent. However, the destabilisation on the treadmill immediately dissipated from the second recovery step, while it lasted

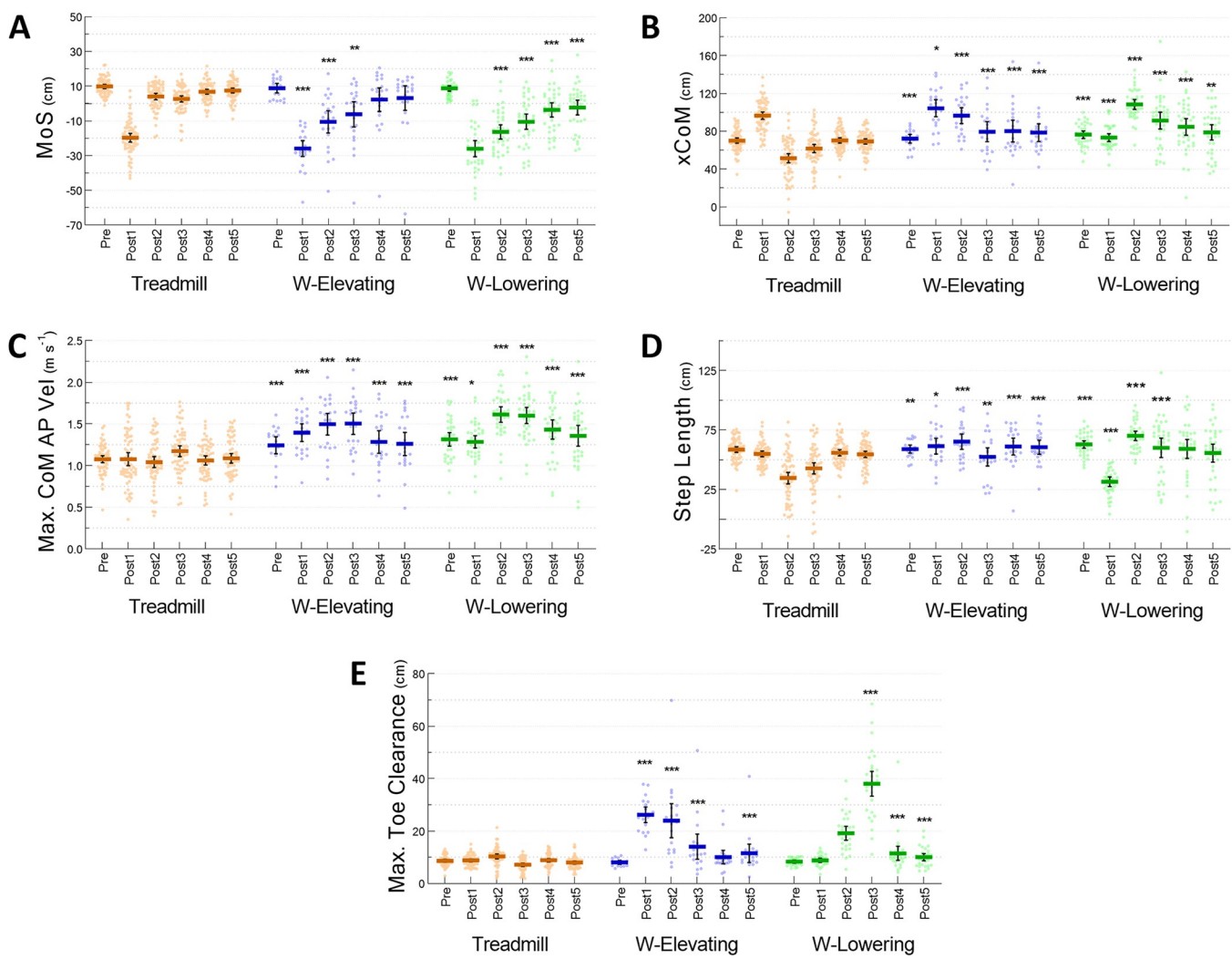

**Fig 2. Kinematic data for one previous (Pre) and five recovery (Post1-5) steps following a treadmill acceleration (orange) and walkway (W) trip inducing elevating (blue) and lowering (green) strategies (*n* = 38).** Presented are (A) margin of stability (MoS), (B) extrapolated centre of mass (xCoM), (C) maximum centre of mass anteroposterior velocity (Max. CoM AP Vel), (D) step length and (E) maximum toe clearance (Max. Toe Clearance). Coloured dots represent individual datapoints. Bold coloured bars and vertical error bars are the average and 95% confidence intervals, respectively. Asterisks represent significant differences (* *P*<0.05, ** *P*<0.01, *** *P*<0.001) between the value of the step of interest on the walkway and the value of the corresponding step on the treadmill.

for multiple recovery steps on the walkway (irrespective of the strategy used) (Fig 2A). Previous studies also showed perturbations with a physical impediment when walking on a treadmill (backward ankle-cable pulls) [28, 35] necessitate more steps to recover stability compared to treadmill belt accelerations alone in older people [36]. The sudden drop and return of MoS around the Post1 step may simply reflect the rapid acceleration and return of treadmill belt speed included in the MoS calculation. Contrary to our second hypothesis, we found greater trunk angles following trips on the walkway (both strategies) than on the treadmill (Fig 4A). Greater trunk flexion angles, forward CoM velocity and displacement and more falls following the walkway trips indicate a greater balance challenge compared the treadmill belt accelerations. The maximum trunk angle on the treadmill was ~20° whereas trunk angles during the walkway trip recovery steps exceeded 30° (Fig 4A). As the head and trunk constitute over 50% of the total body weight [37], greater trunk flexion over the BoS likely contributes to instability

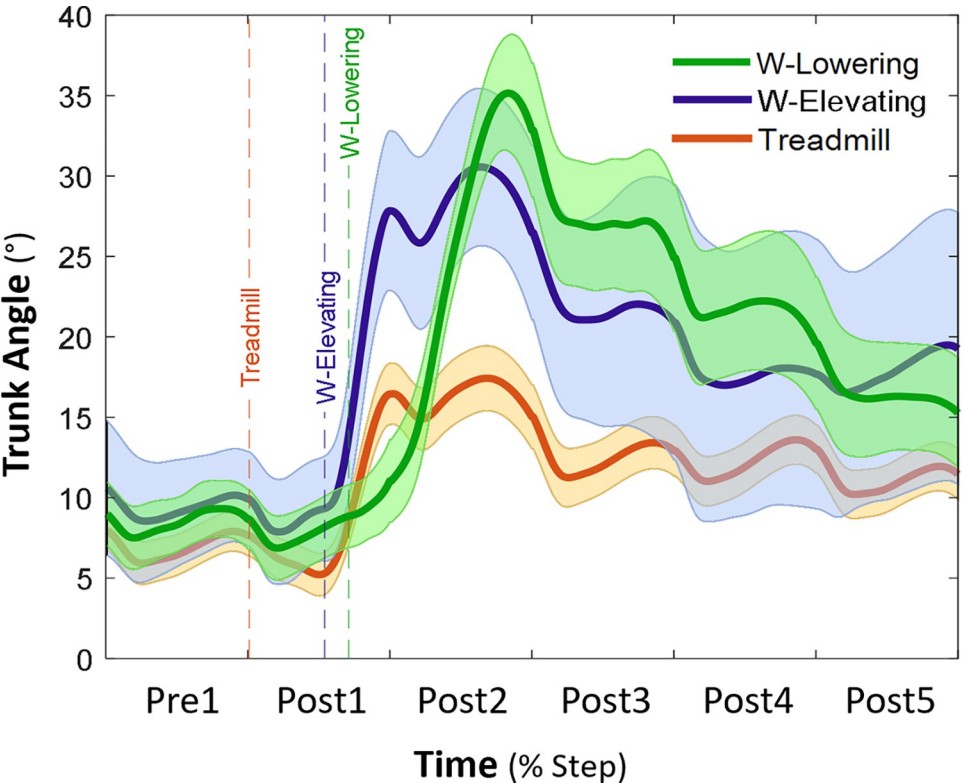

**Fig 3. Trunk angle following a treadmill acceleration (orange) and walkway (W) trip inducing elevating (blue) and lowering (green) strategies (*n* = 38).** Time was normalised to previous (Pre) and recovery (Post1-5) steps. Vertical dashed lines represent perturbation onsets (i.e., a foot contact to the accelerated belt or the trip-board). Bold lines and shaded bands are average and 95% confidence intervals, respectively.

and necessitates greater neuromuscular responses to avoid falling following the walkway trips [6, 20, 38].

### Stepping kinematics

Distinct stepping kinematics were observed between conditions, supporting our third hypothesis. The anteriorly displaced CoM with an increased AP velocity was countered by longer step lengths during both walkway trip recovery strategies (Fig 2). Tripping over an obstacle requires the perturbed foot to be either quickly raised over or brought down for the contralateral foot to sufficiently clear the obstacle with an uncertain dimension [8]. The greatest vertical displacement observed during a recovery step on the treadmill was 10.3±3.5 cm (Post2), which was less than the greatest displacements observed in both the elevating (26.2±6.4 cm [Post1]) and lowering strategies (38.1±12.7 cm [Post3]) (Fig 2E) achieved via greater hip and knee angles in the corresponding recovery steps (Fig 4). The maximum height of recovery steps was considerably higher than the 14 cm trip-board. This may indicate an over-estimation of a novel obstacle to include a "safety margin" to avoid another obstacle collision [15, 39, 40]. In contrast, treadmill belt accelerations without any physical obstructions showed no increase in length or height in recovery steps.

In line with similar studies, our sample of older people had shorter step lengths and higher cadence while walking on the treadmill [41, 42]. This may reflect a "cautious gait" [43] in response to walking on an elevated, moving treadmill surface. However, we confirmed that the

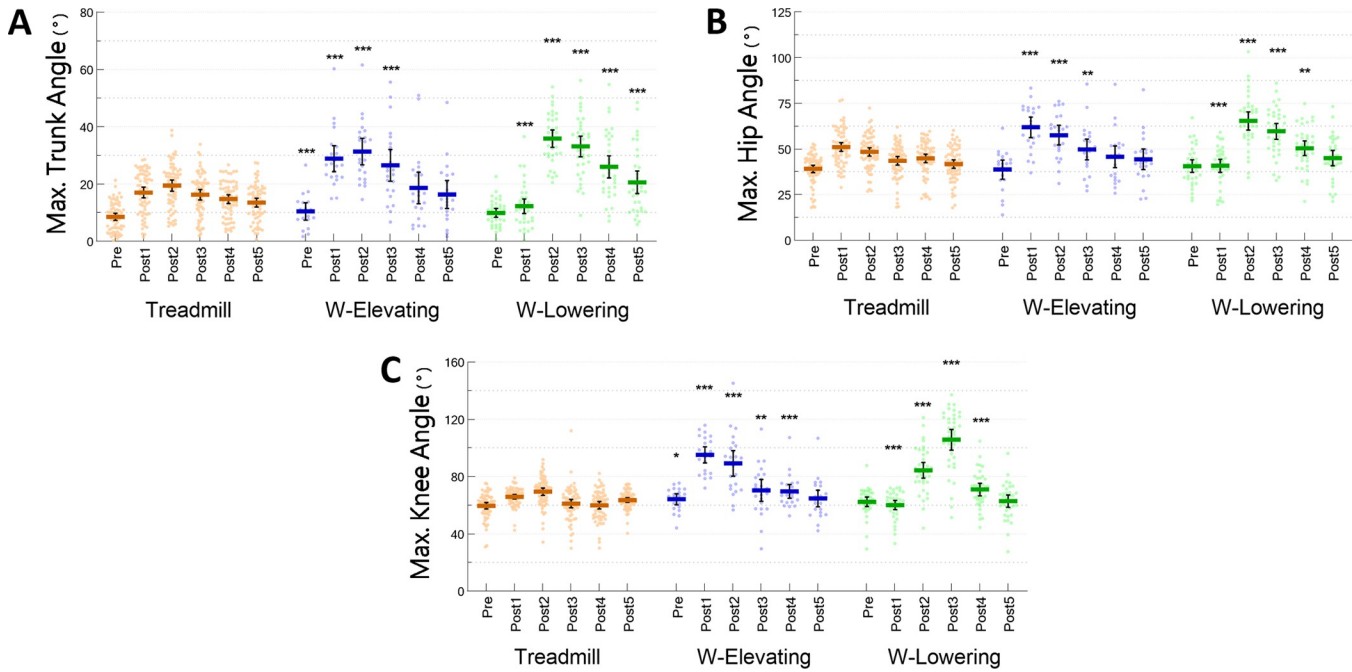

**Fig 4.** Maximum angle of the (A) Trunk, (B) Hip and (C) Knee for one previous (Pre) and five recovery (Post1-5) steps following a treadmill acceleration (orange) and walkway (W) trip inducing elevating (blue) and lowering (green) strategies (*n* = 38). Coloured dots represent individual datapoints. Bold coloured bars and vertical error bars depict the average and 95% confidence intervals, respectively. Asterisks represent significant differences (* *P*<0.05, ** *P*<0.01, *** *P*<0.001) between the value of the step of interest on the walkway and the value of the corresponding step on the treadmill.

margin of stability and gait speed during unperturbed walking on the walkway and treadmill did not differ. Therefore, these gait parameters are unlikely to have contributed significantly to the observed differences in kinematic responses induced by the treadmill belt accelerations and walkway trips.

### Implications for assessment and training using simulated trips

Whilst there is increasing interest in the clinical feasibility of fall prevention training involving simulated trips on a treadmill [24], the current findings suggest that treadmill-induced belt accelerations may not adequately simulate walkway trips. This can explain the lack of transfer from treadmill belt acceleration training to obstacle-induced trips [14]. Treadmill perturbations with higher belt accelerations during gait (e.g., 15 m s$^{-2}$) [30] may increase the initial destabilisation, forward trunk flexion, and potentially recovery step lengths compared to our findings based on a moderate acceleration (8 m s$^{-2}$). However, considering the lack of swing foot obstruction, larger belt accelerations may not induce the key tripping recovery response of heightened recovery steps because there is no need to further elevate the foot. As our middle-aged pilot participant reported muscle strain with a treadmill belt acceleration at 12 m s$^{-2}$, the safety of applying a high acceleration should be carefully considered.

Considering the potential utility of treadmills for perturbation-based training, future research should explore the potential for treadmill belt accelerations to induce responses that better resemble obstacle trips. This may involve increasing the belt acceleration properties, adding a short deceleration prior to a rapid acceleration [30] and accelerating both belts at single stance phase (rather than at a foot-strike). Ankle-cable pulls on a treadmill can physically obstruct the swing foot but repeated exposure to these perturbations lowered toe clearance during steady state walking [44]. Such gait adaptation may minimize the destabilizing effect of

ankle-cable pulls but may not facilitate foot elevation to recover from an obstacle trip. Treadmill training linked with obstacle negotiation appears to modify recovery steps, in that Bieryla et al. found training involving treadmill belt accelerations with a 7.6 cm high obstacle near the feet during stance yielded elevated recovery steps and improved trunk recovery responses following a walkway trip [12]. Furthermore, it has been reported that sudden treadmill start training with a 5 cm high foam obstacle over two weeks reduced the rate of real-life falls in 82 women in the following year [17]. Obstacles appear to be crucial for replicating trip responses, the most common cause of falls in community-dwelling older people [21].

## Study limitations

There are limitations that warrant consideration. First, study participants were relatively healthy and active, thus the findings may not generalise to frailer older people. Second, missing data impacted the walkaway trials more so than treadmill trials, due to greater perturbation impact (markers becoming dislodged), trunk flexion (markers being obscured) and smaller optimum capture volume relative to the long walkway. Third, the additional cognitive demands on participants to standardise their gait on the walkway (e.g., metronome, stepping targets) may have influenced the observed outcomes. Fourth, although MoS and gait speed were similar during unperturbed gait on the walkway and treadmill, shorter step lengths and higher cadence on the treadmill may have affected balance recovery responses. Future research could use the metronome to standardise both cognitive demand and cadence on the treadmill and walkway. Fifth, the treadmill belt accelerations (8 m s$^{-2}$) were delivered at the foot strike (double support stance) form but future studies may investigate different timings (e.g., mid-stance), accelerations (e.g., up to 15 m s$^{-2}$), duration and maximum speed [25]. Future research should investigate if different treadmill acceleration properties or use of additional devices may better simulate kinematic responses induced by obstacle-trips.

## Conclusions

Using maximum acceleration properties considered to be safe for older adults, our study findings suggest that kinematic responses induced by treadmill belt accelerations substantially differ from those induced by trips on a walkway in older people. Destabilisation in MoS dissipated on the treadmill after only one recovery step, but multiple recovery steps were required on the walkway. A greater challenge to balance was also evident by greater trunk flexion, more forward CoM velocity and displacement and falls following walkway trips compared with treadmill accelerations. Furthermore, increased toe clearance and step length following walkway trips were not replicated following treadmill belt accelerations indicating the limitations of treadmill perturbations for simulating walkway obstacle trips. Further research is needed to devise a clinically feasible and kinematically valid method of simulating real-life trips in older people to deliver effective assessment and training strategies.

## Supporting information

**S1 File. Appendix.**
(DOCX)

## Acknowledgments

The authors express their gratitude to Mr. Patrick Sengalrayan for programming the treadmill, Ms. Neesha Krishnan for contributing to the data acquisition, Dr. Kimberley van Schooten for providing feedback on a previously written student thesis and Dr. Peter Humburg for

providing statistical consultation. The authors thank all the participants who voluntarily took part in this study. No financial support was given for this project.

## Author Contributions

**Conceptualization:** Daina L. Sturnieks, Patrick Y. H. Song, Stephen R. Lord, Yoshiro Okubo.

**Data curation:** Dayeon C. Jung, Patrick Y. H. Song, Michael K. Davis, Yoshiro Okubo.

**Formal analysis:** Dayeon C. Jung, Yoshiro Okubo.

**Funding acquisition:** Daina L. Sturnieks, Stephen R. Lord, Yoshiro Okubo.

**Investigation:** Patrick Y. H. Song, Yoshiro Okubo.

**Methodology:** Daina L. Sturnieks, Patrick Y. H. Song, Yoshiro Okubo.

**Project administration:** Daina L. Sturnieks, Kirsty A. McDonald, Patrick Y. H. Song, Yoshiro Okubo.

**Resources:** Daina L. Sturnieks, Yoshiro Okubo.

**Software:** Yoshiro Okubo.

**Supervision:** Daina L. Sturnieks, Yoshiro Okubo.

**Validation:** Yoshiro Okubo.

**Visualization:** Yoshiro Okubo.

**Writing – original draft:** Dayeon C. Jung, Yoshiro Okubo.

**Writing – review & editing:** Dayeon C. Jung, Daina L. Sturnieks, Kirsty A. McDonald, Patrick Y. H. Song, Michael K. Davis, Stephen R. Lord, Yoshiro Okubo.

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
