## [Decision Letter · Decision Letter 0]

2 Apr 2024

PONE-D-23-43759Treadmill belt accelerations may not accurately replicate kinematic responses to tripping on an obstacle in older peoplePLOS ONE

Dear Dr. Okubo,

Thank you for submitting your manuscript to PLOS ONE. After careful consideration, we feel that it has merit but does not fully meet PLOS ONE’s publication criteria as it currently stands. Therefore, we invite you to submit a revised version of the manuscript that addresses the points raised during the review process.

We look forward to receiving your revised manuscript.

Kind regards,

Ryota Sakurai, Ph.D.

Academic Editor

PLOS ONE

Journal Requirements:

2. We noted in your submission details that a portion of your manuscript may have been presented or published elsewhere. "The material within has not been and will not be submitted for publication elsewhere except as a preprint (http://dx.doi.org/10.2139/ssrn.4432392)." Please clarify whether this [conference proceeding or publication] was peer-reviewed and formally published. If this work was previously peer-reviewed and published, in the cover letter please provide the reason that this work does not constitute dual publication and should be included in the current manuscript.

**Additional Editor Comments:**

Since the reviewers have raised reasonable comments, please carefully study their comments.

Reviewers' comments:

Reviewer's Responses to Questions

**Comments to the Author**

1. Is the manuscript technically sound, and do the data support the conclusions?

Reviewer #1: Partly

Reviewer #2: Yes

2. Has the statistical analysis been performed appropriately and rigorously? 

Reviewer #1: Yes

Reviewer #2: Yes

3. Have the authors made all data underlying the findings in their manuscript fully available?

Reviewer #1: Yes

Reviewer #2: Yes

4. Is the manuscript presented in an intelligible fashion and written in standard English?

Reviewer #1: Yes

Reviewer #2: Yes

5. Review Comments to the Author

Reviewer #1: Thank you for the opportunity to peer review. One primary finding is that perturbations from treadmills dissipate rapidly after the second recovery step, whereas those from overground walking persist across multiple recovery steps. In addition, they found treadmill accelerations induced smaller trunk displacement, step lengths, toe clearance and no falls than overground walking. This research is invaluable in terms of addressing the validation of treadmill accelerations for training of trip. This research is valuable for physical therapists. However, please respond to some revisions to improve the clarity of the article.

p.5 Line 5: Please clarify which tasks are being compared in this study. In the introduction, the authors mentioned: "Trips during gait have been simulated on a treadmill by applying blocks to the treadmill surface [24]; [25], backward ankle-cable pulls [26]; [27], rapid belt accelerations on foot strike [28]; [14] or a combination of rapid belt accelerations and decelerations [29] ". However, it remains unclear which specific task was selected for this study, thereby the basis of the hypothesis questionable.

p.5 Line 12: Please specify standard of "validity". I could not understand what would be considered a valid proof in this study.

p.5 Line 15: Similar to the above, the "similar" is ambiguous; please specify further. Focusing on the adequacy of the perturbation magnitude within the surrogate task, as a training stimulus, would enhance clarity.

p.5: Please include a detailed description of the specific kinematic characteristics of the selected task. I could not understand the foundational knowledge that would validate the hypothesis is not well understood.

p.7 Line 8: Please specify the number of perturbation trial in both overground and treadmill walking. Readers cannot understand the number of perturbation trial until they read the analysis.

p.9 Line 20: Please describe this information by gender.

p.11 Line 13: As a reason for the second hypothesis, could you consider whether the relatively small perturbation from the treadmill could be a factor? Discussing the potential for achieving a trunk flexion angle like overground walking by increasing the treadmill's acceleration or perturbation duration would be desirable.

p.12 Line 11: Similar to the above, please discuss whether the treadmill’s perturbation is sufficient. I thought that the relatively small treadmill’s perturbation might did not increase the length nor height of recovery steps.

P12. Line 18: While a perturbation of 15 m/s^2 may indeed be excessively high, is there not a possibility that increasing it beyond 8m/s^2 could induce the treadmill walking more like overground walking? In the introduction, the authors mention the benefits of using a treadmill from a clinical application standpoint; hence, discussing how treadmill training can be optimized would offer valuable clinical implications.

P13. Line 24: If adjusting the treadmill's perturbations can replicate the disturbances experienced during overground walking, then this expression might be considered an overestimation.

Reviewer #2: This paper compared the kinematic responses induced by treadmill belt accelerations and those induced by trips on a walkway in older people. The results clearly indicated that the kinematic responses for the treadmill test substantially differed from those for the walkway test. The experiment was well-designed, and the paper is well-organized and well-written. However, some points need to be clear prior to publication.

Introduction:

1) The problem statement is very clear, but the basis for the derivation of the three hypotheses listed at the end of the introduction is unclear. This point should be stated before stating the objectives.

2) Please explain why the authors think that the acceleration on the supporting leg in treadmill walking can simulate a tripping and trip-induced falls. In this case, the supporting leg moves backward, which seems to simulate a "backward slip" situation, in which the leg slips backward and forward-falls will occur. The rationale for thinking that this can simulate a trip-induced fall rather than a slip-induced forward fall must be presented. Methods:

3) Please show the rationale behind setting the treadmill belt acceleration magnitude and duration (8ms-2 to a maximum of 200% of usual walking speed for 30% of stride time). Can this acceleration conditions simulate the collision of swing foot?

4) What is the definition of AP COM velocity on treadmill walking? Is this an AP COM velocity relative to treadmill belt velocity? Please clarify this.

5) In line 15 on page 8, BOS may be missing after “within the”.

Results

6) Were the elevating and lowering strategy seen in treadmill walking trials? Are these strategies applicable only on walkway trials? Please clarify.

7) The step length was reduced, and the cadence was increased for the treadmill walking. Why did these differences in gait parameters occur, and did the differences in these gait parameters affect the differences in kinematic responses between treadmill walking and over ground walking?

6. PLOS authors have the option to publish the peer review history of their article (what does this mean?). If published, this will include your full peer review and any attached files.

Reviewer #1: No

Reviewer #2: No

---

## [Author Response · Author response to Decision Letter 0]

14 Aug 2024

Dear Reviewers,

We greatly appreciate the time and effort you have dedicated to providing constructive feedback and valuable suggestions. Each of your comments has been carefully considered, and we have implemented revisions to address the concerns raised. Please refer to the response letter where we present a point-by-point response to each comment. We feel that the clarity of the manuscript has improved significantly following the revision and look forward to receiving further comments. 

Sincerely,

Dr Yoshiro Okubo

On behalf of the authors

---

## [Decision Letter · Decision Letter 1]

4 Oct 2024

Treadmill belt accelerations may not accurately replicate kinematic responses to tripping on an obstacle in older people

PONE-D-23-43759R1

Dear Dr. Okubo,

We’re pleased to inform you that your manuscript has been judged scientifically suitable for publication and will be formally accepted for publication once it meets all outstanding technical requirements.

Kind regards,

Ryota Sakurai, Ph.D.

Academic Editor

PLOS ONE

Additional Editor Comments (optional):

Reviewers' comments:

Reviewer's Responses to Questions

**Comments to the Author**

1. If the authors have adequately addressed your comments raised in a previous round of review and you feel that this manuscript is now acceptable for publication, you may indicate that here to bypass the “Comments to the Author” section, enter your conflict of interest statement in the “Confidential to Editor” section, and submit your "Accept" recommendation.

Reviewer #1: All comments have been addressed

Reviewer #2: All comments have been addressed

2. Is the manuscript technically sound, and do the data support the conclusions?

Reviewer #1: Yes

Reviewer #2: Yes

3. Has the statistical analysis been performed appropriately and rigorously? 

Reviewer #1: Yes

Reviewer #2: Yes

4. Have the authors made all data underlying the findings in their manuscript fully available?

Reviewer #1: Yes

Reviewer #2: Yes

5. Is the manuscript presented in an intelligible fashion and written in standard English?

Reviewer #1: Yes

Reviewer #2: Yes

6. Review Comments to the Author

Reviewer #1: I appreciate the careful revisions you have made. After reviewing them, I have found the changes to be sufficient.

Reviewer #2: (No Response)

7. PLOS authors have the option to publish the peer review history of their article (what does this mean?). If published, this will include your full peer review and any attached files.

Reviewer #1: **Yes: **Benio Kibushi

Reviewer #2: No

---

## [Editor Report · Acceptance letter]

6 Nov 2024

PONE-D-23-43759R1 

PLOS ONE

Dear Dr. Okubo, 

I'm pleased to inform you that your manuscript has been deemed suitable for publication in PLOS ONE. Congratulations! Your manuscript is now being handed over to our production team.

Kind regards, 

on behalf of

Dr. Ryota Sakurai 

Academic Editor

PLOS ONE